# Chromatographic Techniques and Pharmacological Analysis as a Quality Control Strategy for *Serjania triquetra* a Traditional Medicinal Plant

**DOI:** 10.3390/ph15101289

**Published:** 2022-10-20

**Authors:** A. Berenice Aguilar-Guadarrama, Guadalupe Yáñez-Ibarra, Martha Edith Cancino-Marentes, Paola González-Ibarra, Rolffy Ortiz-Andrade, Amanda Sánchez-Recillas, Javier-German Rodríguez-Carpena, Yoshajandith Aguirre-Vidal, Irma-Martha Medina-Diaz, Gabriela Ávila-Villarreal

**Affiliations:** 1Centro de Investigaciones Químicas, IICBA, Universidad Autónoma del Estado de Morelos, Av. Universidad 1001, Col. Chamilpa, Cuernavaca 62209, Mexico; 2Centro Nayarita de Innovación y Transferencia de Tecnología A. C. “Unidad Especializada en I+D+i en Calidad de Alimentos y Productos Naturales”, Universidad Autónoma de Nayarit, Tepic 63000, Mexico; 3Unidad Académica de Medicina Humana, Universidad Autónoma de Nayarit, Tepic 63000, Mexico; 4Unidad Académica de Salud Integral, Universidad Autónoma de Nayarit, Tepic 63000, Mexico; 5Laboratorio de Farmacología, Facultad de Química, Universidad Autónoma de Yucatán, Mérida 97069, Mexico; 6Red de Estudios Moleculares Avanzados, Campus III, Instituto de Ecología A.C. (INECOL), Xalapa 91073, Mexico; 7Laboratorio de Contaminación y Toxicología Ambiental, Secretaría de Investigación y Posgrado, Universidad Autónoma de Nayarit, Tepict 63000, Mexico; 8Unidad Académica de Ciencias Químico Biológicas y Farmacéuticas, Universidad Autónoma de Nayarit, Tepic 63000, Mexico

**Keywords:** vasorelaxant activity, traditional medicine, herbal products, ursolic acid, allantoin, NMR, UPLC-MS, CG-MS, blood pressure

## Abstract

*Serjania triquetra* is a medicinal plant widely used in traditional medicine for the treatment of urinary tract diseases, renal affections, and its complications. The population can buy this plant in folk markets as a raw material mixed with several herbal remedies or as a health supplement. On the market, two commercial presentations were found for the vegetal material; one had a bulk appearance and the other was marketed wrapped in cellophane bags (HE*St*-2, HE*St*-3). Nevertheless, the plant has not been exhaustively investigated and quality control techniques have not been developed. This research aimed to realize a phytochemical study using an authentic, freshly collected sample as a reference for *S. triquetra* (HE*St*-1), using the compounds identified. A method for the determination of preliminary chromatographic fingerprinting was developed. Additionally, the vasorelaxant effect from three samples was evaluated with ex vivo rat models. Thus, three hydroalcoholic extracts (HE*St*-1, HE*St*-2, and HE*St*-3) were prepared by maceration. A total of nine compounds were fully identified from HE*St*-1 after the extract was subjected to open-column chromatography. Seven metabolites were detected by gas chromatography, while ursolic acid (UA) and allantoin were isolated and identified using UPLC-MS and NMR, respectively. Three extracts were analyzed for their chromatographic fingerprint by UPLC-MS. Biological activity was explored by ex vivo rat aorta ring model to evaluate vasorelaxant activity. All extracts showed a vasorelaxant effect in a concentration-dependent and endothelium-dependent manner. *S. triquetra* vascular activity may be attributed to UA and allantoin compounds previously described in the literature for this activity.

## 1. Introduction

Hypertension is a multifactorial disease and is a major public health problem due to its high prevalence all around the world. Around 12.8% of the overall of all annual deaths globally arise because of high blood pressure. It is predicted to increase to 1.56 billion adults with hypertension by 2025 and is the most common risk factor leading to hypertensive cardiovascular disease, heart failure, atherosclerotic complications including stroke, coronary heart disease, renal insufficiency, and hypertensive renal disease [1,2]. In this context, these complications and elevated blood pressure are pervasive findings in patients with chronic kidney diseases (CKD). The kidney plays a significant regulatory role in many aspects of blood pressure, particularly in the presence of comorbidities such as diabetes and prevalent cardiovascular disease, contributing greatly to hypertension development as a secondary condition [3]. When kidney function is compromised, the ability of the kidney to maintain balance and blood pressure might be reduced. The main problem for treating elevated blood pressure is due to a silent asymptomatic phase leading to an untreated or uncontrolled stage for most individuals [4].

Herbal medicinal preparations have been used since ancient times as medicines for the treatment of a range of diseases. Increasing worldwide demand for herbal medicinal products requires extensive research, including information about the phytochemical composition related to their clinical curative effects [5,6].

Undoubtedly traditional medicine plays an important role in improving and maintaining health, and for many cultures, it is a fundamental part of their heritage [7,8]. In Mexico, plants are an important element of traditional medicine, and many of them are considered part of the Mexican cultural heritage. Currently, the number of plant species used in Mexico with medicinal attributes varies from 4000 to 6000; however, there are numerous concerns related to the use of herbal medicines because, unlike conventional medications, they are not fully regulated, and most of these species have not been subjected to chemical, toxicological, pharmacological, or clinical investigations [9,10,11,12,13].

*Serjania triquetra* Radlk. (Sapindaceae) is a popular plant used in folk medicine. The *Serjania* genus is native to the neotropics realm: *S. triquetra* is a climbing plant formed by brown bark lianas, commonly known as “Palo de tres costillas”, and is distributed in Mexico and Central America. In Mexican traditional medicine, the aerial parts are used to treat hepatitis, urinary inflammation, infection, kidney stones, kidney diseases, and their complications. *S. triquetra* has been turned into a popular remedy and it can be found in folk markets in its raw form, mixed with several herbal medicine remedies, or as a health supplement. Its e-commerce is also increasing [14,15]. Some pharmacological activities previously reported for *Serjania* genus are anticancer, analgesic, antibacterial, antioxidant, and anti-inflammatory, among others. Some of the compounds that have been identified for this genus are flavonoids, terpenes, steroids, tannins, alkaloids, and saponins [14].

Previous phytochemical studies of the aerial parts have identified stigmasterol, oleanolic acid, morolic acid, hederagenin, and 11α-hidroperoxy-hederagenin [16], with a wide range of pharmacological properties reported for these compounds. However, there is no evidence of exhaustive experimental studies that can demonstrate any of its pharmacological effects [15].

Herbal material is usually a complex phytochemical mixture, thus, identifying its active compounds and their standardization is not an easy task [17]. Usually, the biological activity of medicinal plants cannot be attributed to a single active principle but results from a synergistic interplay of its constituents. Consequently, quality control for fresh or finished herbal preparations is required, either based on phytochemical markers constituents, or its analytical fingerprints, correlated to its pharmacological activity, making it challenging to develop quality control techniques [18].

This research aimed to characterize the phytochemical profile, initially using chromatographic fingerprinting, and to evaluate the vasorelaxant effect of *S. triquetra* in ex vivo rat models.

Major compound isolation led to the identification of nine compounds; ursolic acid (UA) and allantoin were isolated and identified by spectrometric techniques. According to our findings, this plant is rich in UA, a pentacyclic triterpenoid well known for its wide-ranging pharmacological activities.

Three hydroalcoholic extracts were prepared by maceration. *S. triquetra* (HE*St*-1) was an authentic, freshly collected sample and was used as a reference material, HE*St*-1, which was compared with two commercial samples (HE*St*-2, HE*St*3). Reference extract HE*St*-1 was subjected to open-column chromatography for major compound isolation, and nine compounds were fully identified; UA and allantoin were isolated and identified using UPLC-MS and NMR, respectively. Additionally, seven compounds were detected by gas chromatography in non-polar fractions. Elevated blood pressure is a lingering finding in patients with kidney diseases. According to our findings, this plant is rich in UA, a pentacyclic triterpenoid well known for its wide-ranging pharmacological activities [19]. UA showed antioxidant, diuretic, natriuretic, and saluretic activity in hypertension and hemodynamic in in vivo models, according to Somova in 2003 [20]. Furthermore, allantoin, a natural imidazol that is present in the extracts, is reported to elicit agonist activity in imidazoline I-1 receptor (I-1R), and I1 receptors partially mediate the central hypotensive effects of clonidine-like drugs [21]. With these previous findings, we decided to evaluate the extracts using an ex vivo rat aorta ring model to predict biological activity linked to the vasorelaxant effect.

## 2. Results

### 2.1. Extract Preparation and Thin Layer Chromatography (TLC) Comparision for Collected and Comercials Samples of Serjania triquetra

Three extracts were prepared by maceration, and the results are presented in Table 1. Extracts were initially compared by TLC (Appendix A) and no differences were observed between the extracts.

### 2.2. Chemical Composition of Authentic Serjania triquetra Sample

With the intention of developing a standard analytical method to compare commercial products, the chemical characterization of an authentic sample (HE*St*-1) was performed. HE*St*-1 was used to compare commercial samples by chromatographic techniques. The reference extract HE*St*-1 was subjected to open-column chromatography (OCC) for major compound isolation. As a result of OCC, 121 fractions were collected and pooled into 9 groups by chromatographic similarity and analyzed separately. As a result of the chemical analysis, a total of nine compounds were identified. UA and allantoin were isolated and identified using UPLC-MS and NMR, respectively. Additionally, seven compounds were detected by gas chromatography from non-polar fractions.

Compound **1** was obtained as a white powder with a yield of 0.52% from the fraction eluted with ethyl acetate-methanol in a ratio of 8:2 from *Rn10* in OCC. The precipitate was insoluble in methanol, and the presence of a single compound was observed using TLC. Thus, NMR experiments were obtained, and the list of experimental chemical shifts is shown in Table 2. The analysis of data obtained from the experiments in 1D (^1^H NMR and ^13^C NMR), shows that this compound corresponds to (2,5-dioxoimidazolidin-4-yl) urea, commonly known as allantoin. This was confirmed with spectroscopy literature data, as shown by Rasheed et al. [22].

Moreover, *Rn11-12* in OCC resulted in the presence of a second compound. The compound was a powder which was soluble in methanol, identified as UA (IUPAC name (1*S*,2*R*,4a*S*,6aR,6aS,6bR,8aR,10S,12aR,14bS)-10-hydroxy-1,2,6a,6b,9,9,12a-heptamethyl-2,3,4,5,6,6a,7,8,8a,10,11,12,13,14b-tetradecahydro-1H-picene-4a-carboxylic acid). Elution in TLC showed the characteristic purple coloration of the pentacyclic triterpenes. The result was compared with an authentic standard sample of UA using TLC. Both samples had a similar retention factor, and the presence of UA was corroborated by ultra-performance liquid chromatography coupled with mass spectrometry (UPLC-MS) using a selected ion recording experiment (SIR), where [M-H]^−^ 455.72 and 455.38 Da was found in the authentic UA and the sample, respectively.

Additionally, seven compounds (Figure 1) from the non-polar fractions (groups *Rn01*-*Rn05*) of the EHA*St*-1 were identified using gas chromatography coupled with mass spectrometry (GC-MS). The five compounds eluted in 100% dichloromethane fractions (*Rn01*, compounds **3–7**, 0.15%) and two compounds in the fractions eluted with dichloromethane-ethyl acetate in a ratio of 7:3 (*Rn05*, compounds **8** and **9**, 0.54%). Table 3 shows the chemical composition and the content in the percentage of compounds analyzed by GC-MS in the non-polar fractions. After GC-MS identification, these compounds were further confirmed using targeted UPLC-MS in SIR mode.

### 2.3. Comparison of Authentic and Commercial Botanical Samples: Botanical Reference Fingerprint Development by UPLC-MS as Quality Control Suggested Technique

A UPLC-ESI-MS method was developed with the purpose of creating a rapid and easy quality control technique for *S. triquetra* samples. For each hydroalcoholic sample extract, both authentic (HE*St*-1) and commercial samples (HE*St*-2, HE*St*3), a full mass scan and a targeted SIR experiment were conducted. For each SIR experiment, nine different channels were selected, one channel for each mass of the nine identified compounds. The results are shown in Figure 2. The signal intensity in samples HE*St*-1, HE*St*-2, and HE*St*-3 is different, where the fresh plant HE*St*-1 provided the highest intensity compared to HE*St*-2 and HE*St*-3. Figure 2 shows the variation of signal intensities for the three herbal samples prepared by the same conditions and analyzed in the same UPLC-MS method, as indicated in the total ion chromatogram and the SIR experiment for each compound. These differences in signal intensity between the samples could infer the storage condition of the herbal materials. For the herbal material, bulk-marketed HE*St*-2 and cellophane-marketed HE*St*-3 in Figure 2A, we can observe that the lowest signal intensity was for the bulk-marketed samples (HE*St*-2) compared to the freshly collected material. Also, we can discern that herbal material cellophane preserve shows a major intensity of signals than bulk marketed material but lower than freshly collected material. These findings support the importance of good agricultural and commercialization practices for herbal medicinal products.

### 2.4. Pharmacological Evaluation (Ex Vivo Model)

Evaluation of all HE*St* extracts by ex vivo model has shown that it produces a vasorelaxant effect in aortic rings pre-contracted by noradrenaline. In this study, it was found to induce a significant relaxant effect in a concentration-dependent and endothelium-dependent manner (Figure 3). All extracts from *S. triquetra* were more potent and less efficient than carbachol and less potent and efficient than nifedipine, both used as positive controls in the presence and absence of endothelium, respectively.

## 3. Discussion

The increasing requirement for herbal medicines certainly leads to providing quality controls for herbal raw materials. Standardization relating to herbal drugs arises from the complex composition of traditional medicine that is used in the form of whole plants or plant parts [7]. With the objective of assuring the reproducible quality control methods of any herbal remedy, the proper collection and identification of the authentic herbal material are essential; in addition to this, performing phytochemical studies to develop these quality control strategies is crucial in the process. Developing quality control strategies for herbal remedies from the starting material needs to consider several factors. For this project, we linked the phytochemical study with an ex vivo model, indicating the potential activity in all samples, as shown in Figure 3, when UA and allantoin are present, the vasorelaxant effect is preserved in the three extracts [18,21]. Furthermore, preliminary chromatographic fingerprinting was developed by ultra-performance liquid chromatography coupled with mass spectrometry as a quality control approach.

Persistent elevated blood pressure can potentially lead to kidney damage and is a lingering finding in patients with kidney diseases. According to our findings, *S. triquetra* is rich in UA, a pentacyclic triterpenoid well known for its wide-ranging pharmacological activities. UA showed antioxidant, diuretic, natriuretic, and saluretic activity in hypertension and hemodynamic in vivo models, according to Somova in 2003 [20]. Furthermore, allantoin, a natural imidazol that is present in the extracts, is reported to elicit agonist activity in imidazoline I-1 receptor (I-1R), while I1 receptors partially mediate the central hypotensive effects of clonidine-like drugs. With these previous findings, we decided to evaluate the extracts using an ex vivo rat aorta ring model to predict biological activity linked to the vasorelaxant effect. As shown in Figure 3, the vasorelaxant effect is preserved in the three extracts [21].

Thus, we hypothesize that all these named compounds in *S. triquetra* extracts effects contributed to a reduction in peripheral vascular resistance, leading to a significant hypotensive effect. These results are novel for *S. triquetra.*

## 4. Materials and Methods

### 4.1. Plant Material

For the development of a reference, stems of *S. triquetra* were collected in Santiago Ixcuintla, Nayarit, Mexico, on 21 November 2017 (GPS coordinates: 104°59′47.1″ W, 21°44′27.0″ N). The botanical identification was made by Ana María Hanan Alipi (Universidad Autónoma de Nayarit), and the voucher specimen was deposited in the Herbarium of the Universidad Autónoma de Nayarit, Tepic, Nayarit (Herbarium UAN 1751). Additionally, two samples were purchased commercially in two different presentations, one in bulk form to prepare HESt-2 and the other in cellophane bags to prepare HE*St*-3 extracts.

### 4.2. Extract Preparation, Extraction, and Isolation

Three hydroalcoholic extracts were prepared by maceration. An authentic freshly collected sample was used as a reference for *S. triquetra* (HE*St*-1) and was compared with two commercial samples (HE*St*-2, HE*St-*3). Reference extract HE*St*-1 was subjected to open-column chromatography.

The dry stems were pulverized using a mill (Romer mill series II^®^, model 4679737, Union, MO, USA). The milled sample was extracted with a hydroalcoholic solution (85%) for 72 h at room temperature after extraction and filtered using filter paper Whatman No. 1 (Whatman^TM^, Cat No. 1001.110, Maidstone, UK). The solvent extract was removed by a rotary evaporator (Buchi^®^, Gwangmyeong, Korea), as established in the General Method of Analysis MGA-FH 0070, extractable material of the Herbal Pharmacopoeia of the United Mexican States [15]. For extract preparation, 850 g was used for HESt-1, 500 g for HE*St*-2, and 210 g for HE*St*-3.

For chromatographic fractionation, 8 g of the complete extract from authentic fresh collected plant (HE*St*-1) was subjected to OCC in which the silica gel 60 (0.063–0.200 mm Mesh) was packed in a glass column (45 mm × 500 mm), using a dichloromethane–ethyl acetate–methanol–water elution gradient system and mixes of these solvents. Eluted fractions were monitored by thin-layer chromatography using aluminum sheets of TLC silica gel 60 F_254_.

### 4.3. Nuclear Resonance Magnetic (NMR)

A Varian Mercury 200 equipment at 200 and 50 MHz frequencies were used to obtain the ^1^H and ^13^C experiments, respectively. Samples were dissolved in deuterated dimethyl sulfoxide-*d*_6_ (DMSO-*d*_6_) and TMS as reference obtained from Sigma-Aldrich (St. Louis, MO, USA).

### 4.4. Development of Quality Control Methodology by Fingerprinting Analysis Using Ultra-Performance Liquid Chromatography Coupled with Mass Spectrometry (UPLC-MS)

Chromatographic separation was performed using ACQUITY UPLC H-Class Bio System (Waters Corp., Milford, MA, USA). The separation was conducted using an ACQUITY UPLC^®^ HSS T3 130 Å column (1.8 µm, 2.1 mm × 50 mm, Waters^®^, Milford, MA, USA), column temperature 35 °C with an isocratic elution of 20% A to 80% B in 10 min using a binary system consisting of ammonium hydroxide (0.05%) in water (A) and acetonitrile (B). 3 μL of the samples at 100 ppm concentration were injected with a flow rate of 0.4 mL/min; acetonitrile was used as a blank solvent.

Detection was performed using ACQUITY QDa detector mass spectrometer (Waters Corp., Milford, MA, USA) with an electrospray ionization interface (ESI); the voltage of the capillary was set to −2 kV for the negative ion mode (ESI-). The data was processed using Waters Empower™ 3 software (Waters Corp., Milford, MA, USA). A Mass scan acquisition was programmed at 50 to 1250 Da, and SIR for each known mass was selected.

### 4.5. Gas Chromatography Coupled with Mass Spectrometry (GC-MS)

An Agilent plus equipment coupled to a mass detector (Agilent 5379N 30-5550; electron impact ionization mode at 70 eV) with an HP5-MS capillary column (30 m × 0.225 mm × 0.25 μm) was used to acquire the gas chromatography–mass spectrometry (GC–MS) experiments. Samples (1 μL) were injected in splitless mode (50 °C, 0 min; 2 °C/min; 285 °C, 20 min); helium was the carrier gas at a flow rate 1 mL/min. The database used for comparison of the results was NIST-MS version 1.7a.

### 4.6. Ex Vivo Pharmacological Evaluation

#### 4.6.1. Chemicals and Drugs

Carbamoyl-choline (carbachol: CCH), norepinephrine: HCl (NE), and nifedipine were purchased from Sigma-Aldrich Co., Ltd. (St. Louis, MO, USA). All other reagents were analytical grade from local sources. Stock solutions of extracts were prepared with distilled water on the same day of experimentation.

#### 4.6.2. Animals

Adult male Wistar rats (250–300 g bodyweight) were obtained from the Animal House of Universidad Juarez Autónoma de Tabasco (UJAT), Mexico. Animals were housed in polycarbonate cages and maintained under standard laboratory conditions (12-h light/dark cycle, at a temperature of 25 ± 2 °C, and with a humidity of 45–65%) and were fed with standard rodent diet and water ad libitum. All animal procedures were conducted in accordance with our Federal Regulations for Animal Experimentation and Care (SADER, NOM-062-ZOO-1999, México) and approved by the Institutional Animal Care and Use Committee (Register: CEBN/13/2018). All experiments were carried out using six animals per group. All study animals were sacrificed by cervical dislocation after deep anesthesia with ether.

#### 4.6.3. General Procedures

Rats were sacrificed by cervical dislocation after deep anesthesia, and aorta tissue dissection was carried out to remove the thoracic aorta. This was cleaned from adjacent and connective nerves and then cut into rings of 3 mm in length. In addition, for some aortic rings, the endothelium layer was gently removed by manual procedures. Then, the tissue sections were assembled using stainless steel hooks under an optimal tension of 3 g. After, tissues were allowed to stabilize for 20 min at 37 °C in Krebs–Henseleit physiological solution (KHS; composition in mM: NaCl, 118; KCl, 4.7; KH_2_PO_4_, 1.2; MgSO_4_:H_2_O, 1.2; CaCl_2_ H_2_O, 2.5; NaHCO_3_, 25; EDTA, 0.026, and glucose, 11.1; pH, 7.4) constantly bubbled with an O_2_:CO_2_ (95:5) mixture. Tension changes were recorded by Grass-FT03 force transducers (Astro Med, West Warwick, RI, USA) connected to an MP150 analyzer (BIOPAC 4.1 Instruments, Santa Barbara, CA, USA) as described previously by Sánchez-Recillas et al., 2019 [23].

#### 4.6.4. Vasorelaxant Activity

After the stabilization period, each aorta ring was stimulated with norepinephrine (NE [0.1 μM]) for 15 min, then washed with fresh KHS, and allowed to stabilize for 15 min. This procedure was repeated thrice. The absence of functional endothelium was confirmed by the lack (>50%) of the relaxant response induced by carbachol (CCH [1 μM]) in the last contraction with NE prior to washing with fresh KHS to assess viability.

When contraction reached a plateau, *S. triquetra* extracts (0.0011 to 100 μg/mL), vehicle (100% of DMSO; maximum concentration), and positive controls (CCH; a cholinergic antagonist: 6.28 × 10^−4^ to 1.83 μg/mL or nifedipine: a calcium channel blocker: 3.89 × 10^−5^ to 3.46 μg/mL) were added to the bath in cumulative concentrations. Then, concentration-response curves were obtained.

### 4.7. Data Analysis

The vasorelaxant activity of extracts and positive controls were determined by comparing the muscular tone of the contraction before and after the application of the test materials. The muscular tone was calculated from the tracings using Acknowledge 4.1 software. Concentration-response curves were plotted and fitted with Origin Lab version 8.0 software (Microcal^®^, Northampton, MA, USA).

#### Statistics

The ex vivo model experimental results were expressed as mean ± S.E.M of six experiments. Statistical analysis was done by two-way analysis of variance (ANOVA) with the Bonferroni test using the Origin software. *p* value less than 0.05 was considered statistically significant.

## 5. Conclusions

The results show that the hydroalcoholic extract of *S. triquetra* has a concentration-dependent and endothelium-dependent vasorelaxant effect, and in addition, UA was identified as one of the major compounds of the extract. There are several reports of the vasorelaxant effect of the UA in an endothelium-dependent manner and its possible mechanisms of action involving nitric oxide release in functional experiments [20,24,25]. Therefore, we can conclude that part of the vasorelaxant effect observed in all extracts may be due to UA and the additive effect of allantoin [21], which we can observe is one of the major compounds present in chromatographic analysis with a major intensity for all samples, without ruling out the future evaluation of the other compounds in the elucidation process isolated from the extract.

To our knowledge, this is the first pharmacological report regarding the vasorelaxant effect of this plant and of the compounds reported in this work that may be linked to its traditional use for chronic kidney disease complications. This is one of the most common conditions leading to secondary hypertension due to a range of causes, such as sodium retention with volume overload, renin angiotensin aldosterone axis activation, increased activity of the sympathetic nervous system, and decreased production of endogenous endothelial vasodilators, all of which are countered by the plant’s consumption [26,27]. Many reports of the renoprotective effects of UA have been described via oxidative stress, inflammation, and inhibition of the activities of the signal transducer and activator of transcription (STAT)3/nuclear factor (NF)-κB signaling pathway. Previous studies demonstrated that pre-treatment with UA significantly increased renal functioning and attenuated increases in serum angiotensin II levels in rats [28]; thus, it could be linked to the ex vivo experiments findings and the traditional use of this plant.

Many analytical methods for identification or quantification can be used for quality control purposes, such as gas chromatography with flame ionization detector (GC-FID), gas chromatography mass spectrometry (GC-MS), non-aqueous capillary electrophoresis (NACE), high-performance liquid chromatography with UV (HPLC-UV) and with diode array detector (HPLC-DAD), high-performance thin layer chromatography (HPTLC), and high-performance liquid chromatography–diode array detector–tandem mass spectrometry (UPLC-DAD-MS/MS). The implementation of techniques by UPLC-MS has become popular in recent years due to known advantages such as high sensitivity, fastness, and extensive applications. Furthermore, MS analysis provides more structural information on the chemical profile than the conventional chromatographic coupled to light detectors (HPLC-UV or HPLC-DAD), allowing for more confident identification. For quantitative MS analysis, the method of choice relies on tandem systems, mainly composed of triple quadrupole detectors, since higher sensitivity is achieved by monitoring specific ion transitions (the so-called multiple reaction monitoring MRM) [29,30]. This approach has been successfully applied in many quality control applications. Alternatively, it is possible to filter the m/z from a target ion by a single MS detector. Indeed, this method, known as SIM (selected ion monitoring), is not as sensitive as MRM, however, its cost-benefit has positioned it as a technique with potential for many purposes. Sensitivity, fastness, and broad applications [31]. With these results, we demonstrate that our method, developed for targeted ion identification, allows a rapid, sensitive, and reliable tool implementation for quality control of herbal products when previously, a full chemical and botanical identification has been required.

## Figures and Tables

**Figure 1 pharmaceuticals-15-01289-f001:**
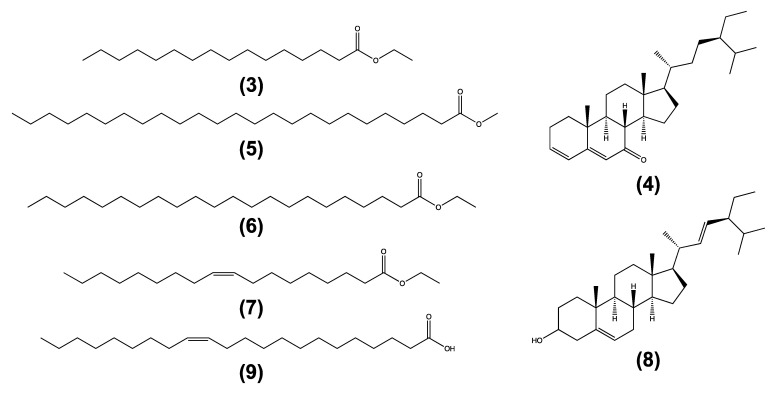
Compounds identified by comparison with NIST database. Ethyl palmitate (**3**) (25.773%), stigmasta-3,5-dien-7-one (**4**) (13.927%), methyl pentacosanoate (**5**) (10.752%), ethyl docosanoate (**6**) (8.640%), ethyl oleate (**7**) (4.973%), stigmasta-5,22-dien-3-ol (**8**) (9.968%), and erucic acid (**9**) (1.434%).

**Figure 2 pharmaceuticals-15-01289-f002:**
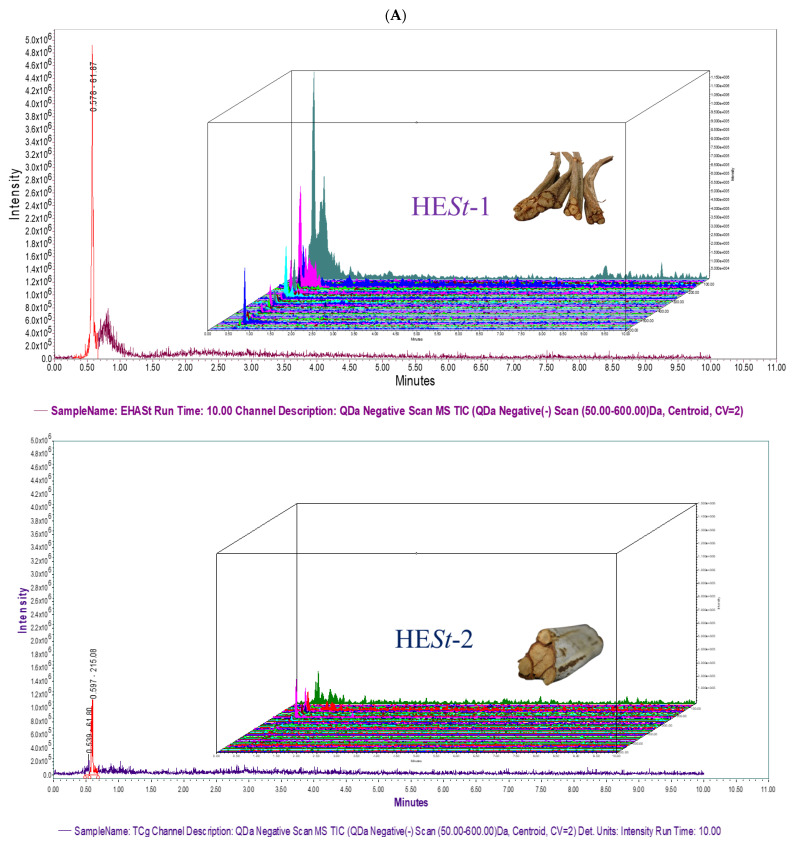
Fingerprinting development by UPLC-MS experiments for HESt-1, HESt-2, and HESt-3, all in negative mode (ESI-). (**A**) Total ion chromatogram: exploratory mass scan from 50–600 Da, the X-axis represents time, and Y-axis represents signal intensity combined with 3D Chromatogram. X-axis represents time, Y-axis represents signal intensity, and Z-axis represents mass in Da. For HE*St*-1, HE*St*-2, HE*St*-3, (**B**) SIR experiments: For identified compounds **1** allantoin 157.12 Da, **2** ethyl palmitate 283.50 Da, **3** ethyl oleate 309.50 Da, **4** erucic acid 337.60 Da, **5** ethyl docosanoate 367.60 Da, **6** methyl pentacosanoate 395.70 Da, **7** stigmasta-3,5-dien-7-one 409.70 Da, **8** stigmasta-5,22-dien-3-ol 411.70 Da, and **9** ursolic acid 455.41 Da. Further details can be found in the Appendix A.

**Figure 3 pharmaceuticals-15-01289-f003:**
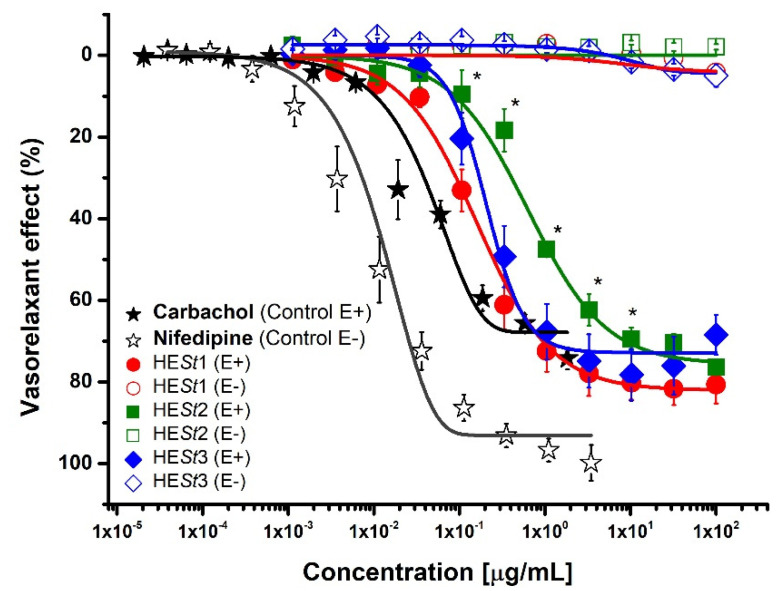
The vasorelaxant effect induced by *S. triquetra* extracts on aortic rings pre-contracted with NE [0.1 µM]. Results are expressed as the mean ± S.E.M of six experiments. (*) Indicates significance on comparison between all extracts (*p*< 0.05).

**Table 1 pharmaceuticals-15-01289-t001:** Extraction yields obtained from stems of *Serjania triquetra*.

Code	Dry Material (g)	Extract %	Description
HE*St-1*	850	14.21	Freshly collected material
HE*St-2*	500	6.63	Herbal remedy bulk marketed
HE*St-3*	210	9.84	Tea marketed in cellophane bags

**Table 2 pharmaceuticals-15-01289-t002:** ^1^H and ^13^C NMR chemical shifts (δ, ppm) of allantoin in DMSO-_d6_.

Position	δ C	δ H
1	-	10.58
2	157.3	-
3	-	6.91
4	62.90	8.07
5	174.0	
6	-	5.23
7	157.90	
8		5.81

**Table 3 pharmaceuticals-15-01289-t003:** Chemical composition of groups 1 and 5 obtained from OCC fractionation of the hydroalcoholic extract of *S. triquetra*.

Compound	Group	RT ^†^	RI ^a^	RI ^b^	Compound Name	Essential Oil (%)
3	1	65.592	1959	1975	Ethyl palmitate	25.773
4	1	115.652	3311	2718	Stigmasta-3,5-dien-7-one	13.927
5	1	97.542	2745	2822	Methyl pentacosanoate	10.752
6	1	90.415	2549	2576	Ethyl docosanoate	8.640
7	1	73.225	2126	2175	Ethyl oleate	4.973
8	5	111.389	3169	3248	Stigmasta-5,22-dien-3-ol	9.968
9	5	78.743	2254	2546	Erucic acid	1.434

RT ^†^ Retention time in minutes; RI ^a^ calculated, retention indices; RI ^b^ reported.

## Data Availability

Data is contained within the article and Appendix A.

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
