# Peer review of "Chromatographic Techniques and Pharmacological Analysis as a Quality Control Strategy for Serjania triquetra a Traditional Medicinal Plant"

_pharmaceuticals, 2022, doi:10.3390/ph15101289_

Round 1
Reviewer 1 Report
Thanks Authors for submitting an interesting research, related to use of plants as natural medicamenta. Elevated blood pressure is potentially serious and can lead to kidney damage. The plant aim of this project is widely used to treat among others, kidney disease and its complications. Authors hypothesized that all these named compounds in S. triquetra extracts effects contributed to a reduction in peripheral vascular resistance, leading to a significant hypotensive effect. These results are novel for S. triquetra.
Interesting interdisciplinary contribution (analytic and pathophysiology) to determination of S. triquetra effects by using an animal model.
Analytical approach are quite and clearly indicated (Development of quality control methodology by Fingerprinting analysis using Ultra-performance liquid chromatography coupled with mass spectrometry (UPLC-MS); Gas chromatography coupled with mass spectrometry (GC-MS)
The results show that the hydroalcoholic extract of S. triquetra has a concentration-dependent and endothelium-dependent vasorelaxant effect, in addition to that UA was identified as one of the major compounds of the extract. There are several reports of the vasorelaxant effect of the UA in an endothelium-dependent manner and its possible mechanisms of action, so Authors can conclude that part of the vasorelaxant effect observed in the HESt may be due to UA and the additive effect of allantoin which they can observe is one of the major compounds present in chromatographic analysis, also with the major intensity for all samples. Without ruling out the future evaluation of the other compounds in the elucidation process isolated from the extract.
Author Response
We appreciate the comments, english is improved.
Reviewer 2 Report
The overall flow of the manuscript is poor as it does not clearly reflect the description in the title. Major revisions will be required and here are some of the comments and suggestions:
1. The abstract is not clear. The problem statement and significance of this work could be written with increased clarity.
2. All figures must be improved with increased font size. It is difficult to read.
3. The authors identified 9 unique compounds from the extracts for the chromatographic analysis. The quantity and ratio of each compound in the extract is not clearly presented in the results.
4. For the pharmacological analysis, did the authors analyse which of the identified compounds to be used as markers is the active component responsible for the vasodilatory effects? This should be a key finding for this paper instead of showing the vasodilatory effect of the entire extract as a whole.
5. Instead of just having an ex vivo study for the pharmacological analysis, could the authors supplement the pharmacological analysis with an in vitro assay? It will probably be good that the assay is of high-throughput study and in support to the ex vivo animal studies shown.
6. The discussion and conclusion sections require major corrections. Especially for the conclusion, it is not clear to the readers what is the significance of this study and how does it help advance in this field/ area of research.
Author Response
Thank you for you comments. Our reply is attached in word document.

Reviewer 3 Report
The manuscript deals with quality control of Serjania triquetra and it is suitable for publications after English-editing to improve the quality of the manuscript.
Author Response
Thank you for your observations. The English-editing will be addressed as requested.
Round 2
Reviewer 2 Report
Authors did not address my concerns adequately.
The main figures should be made of a higher resolution and of better clarity. It is unsatisfactory to rely solely on the supplementary information provided.
The test of Urosolic acid in the ex vivo model is necessary and is a key point in this manuscript.
Author Response
We thank the reviewer for their time, expertise, and effort in conducting constructive reviews that have improved the quality of this paper. Hopefully, with these changes, the paper will be acceptable for publication.

Round 3
Reviewer 2 Report
The authors did not address the major concerns raised in the first and second round of review. Noted that they provided evidence from previous reports of the vasodilatory activity of UA. So, if UA was shown to have vasodilatory effects, what about the other compounds present in the extract? If UA is absent in the extract will there still be any vasodilation effects? If not, what are the effects of the other identified compounds? Are there any high-throughput assays to showcase the identification and activity of these identified compounds?
Author Response
We thank the reviewers for their time, expertise, and effort in conducting constructive review that have improved the quality of this paper. We hope that, with these responses, the paper will be acceptable for publication
Please find attached in word document our responses.
